# Dynamic Taylor Convolutional Neural Network for Few-Shot Point Cloud Semantic Segmentation

## Abstract

Few-shot point cloud semantic segmentation remains a challenge in the field of computer vision due to the limitations of the pre-training learning paradigm and insufficient local geometric structure representation. To address this issue, we propose a novel pre-training-free Dynamic Taylor Convolutional Neural Network, called DyTaylorCNN ingeniously, which combines the potential of the Taylor series in local structure representation with the flexibility and adaptability of dynamic convolutions. The core of DyTaylorCNN lies in two innovative components: the Dynamic Taylor Convolution (DyTaylorConv) and the Interactive Prototype Refinement (IPR) Module. Inspired by the Taylor series and dynamic convolution, DyTaylorConv performs local structure fitting by collaborating between the Low-order Convolution (LoConv) and the Dynamic High-order Convolution (DyHiConv). LoConv is designed based on position encoding, focusing on extracting the basic geometric information of point clouds, while DyHiConv adaptively models complex local geometric features by learning spatial priors to generate dynamic weights. Moreover, the IPR Module effectively reduces the domain distribution gap by learning fine-grained prototype features, further enhancing the model's generalization capability. Experimental results on multiple benchmark datasets demonstrate that the proposed DyTaylorCNN significantly outperforms current state-of-the-art methods.

## 1 Introduction

In recent years, with the rapid development of 3D sensing technology, point cloud data has been increasingly applied in fields such as autonomous driving (Zhao et al., 2023; Chib & Singh, 2023), robotics (Soori et al., 2023; Goel & Gupta, 2020), and augmented reality (Devagiri et al., 2022; Sereno et al., 2020). As a key task in 3D scene understanding, point cloud semantic segmentation (Lai et al., 2022b) plays a crucial role in advancing these applications. However, acquiring large-scale, high-quality annotated point cloud data often requires substantial time and human resources, severely limiting the practical application of traditional deep learning methods.

To alleviate the problem of data scarcity, researchers (Li et al., 2024; Xiong et al., 2024) have begun to focus on few-shot learning strategies for point cloud segmentation tasks. These methods aim to effectively segment new categories using only a small number of annotated samples. Among them, Zhao et al. (Zhao et al., 2021b) first introduced few-shot learning to point cloud semantic segmentation and proposed the AttMPTI method based on pre-trained DGCNN (Wang et al., 2019). Subsequent works (Mao et al., 2022; Lai et al., 2022a; Zhu et al., 2023) further improved feature extraction and prototype generation strategies, enhancing model performance to some extent.

Nevertheless, applying few-shot learning to point cloud semantic segmentation still faces numerous challenges. Firstly, as shown in Fig. 1a, existing methods generally rely on pre-training learning paradigms, which not only increase time and computational costs but may also lead to severe domain shift problems when facing unseen categories. Secondly, the irregularity and sparsity of point cloud data make it a formidable task to effectively capture local geometric structures, especially in few-shot scenarios where this problem becomes more

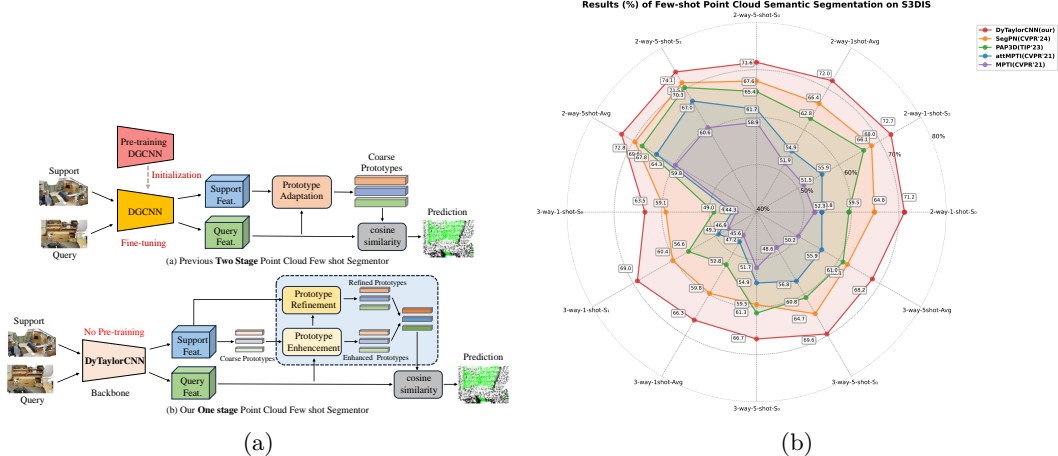

(a)                                          (b)

Figure 1: (a) Top: Most existing methods are based on fine-tuning a pre-trained DGCNN, followed by using query features to guide and align the prototype features. This two-stage approach is not only time-consuming but also overlooks the importance of local structure representation. Bottom: We propose a new DyTaylorCNN that requires no pre-training and possesses strong local structure representation capabilities. Additionally, we design an IPR module that effectively aligns query features with prototype features. (b) Our method achieves the best results over state-of-the-art methods.

pronounced. Lastly, due to sample scarcity, the feature distribution in the support set may significantly differ from that in the query set, affecting segmentation performance.

To address these issues, as illustrated in Fig. 1a, this paper proposes a novel Dynamic Taylor Convolutional Network (DyTaylorCNN) for few-shot point cloud semantic segmentation that requires no pre-training. Firstly, to effectively solve the problem of local structure representation in point clouds, we designed the Dynamic Taylor Convolution (DyTaylorConv) inspired by Taylor series (Rudin et al., 1964) and dynamic convolution (Yang et al., 2019). This convolution views local structure representation as a polynomial fitting problem, using Low-order Convolution (LoConv) based on position encoding to fit the flat parts of local structures, and High-order Convolution (HiConv) to construct multiple high-dimensional geometries in local neighborhoods to fit edges and details, thus more accurately capturing subtle changes in local geometric information. Secondly, to effectively address the domain difference between support and query sets, we designed the novel Interactive Prototype Refining (IPR) Module. This module first learns coarse semantic category prototypes from the support set, then enhances coarse prototypes by learning the spatial distribution of support and query sets, and learns their common semantic space to generate more refined prototype feature representations. This not only effectively reduces inter-domain distribution gaps but also further improves the model's generalization ability in few-shot scenarios. As shown in Fig. 1b, our method achieves significant results in few-shot settings.

The main contributions of this paper can be summarized as follows:

1. We propose an novel DyTaylorCNN for few-shot point cloud semantic segmentation, which achieves excellent performance without time-consuming pre-training processes.

2. We design an innovative DyTaylorConv, which ingeniously combines the expressive power of Taylor series and dynamic convolution, significantly enhancing the model's ability to represent local geometric structures of point clouds.

3. We introduce a IPR Module, which employs a coarse-to-fine learning strategy to generate fine-grained prototype features and effectively bridges the domain gap between support and query sets.

## 2 Related Work

**Point Cloud Semantic Segmentation**. Point cloud semantic segmentation (Wang et al., 2022; Wu et al., 2022) is a crucial task in 3D scene understanding that has witnessed significant advancements in recent years. Pioneering works such as PointNet (Qi et al., 2017a) and PointNet++ (Qi et al., 2017b) established the foundation by directly processing point cloud data through multi-layer perceptrons (MLPs). Subsequent research introduced innovative methods leveraging graph convolution, attention mechanisms, and multi-modal approaches. For instance, DGCNN (Wang et al., 2019) proposed the EdgeConv operation to capture inter-point relationships via dynamically constructed local graphs. Point Transformer (Zhao et al., 2021a) and its variants like Stratified Transformer (Lai et al., 2022b) and Fast Point Transformer (Park et al., 2022) incorporated self-attention mechanisms to effectively model long-range dependencies and improve processing efficiency. RandLA-Net (Hu et al., 2020) achieved efficient large-scale point cloud segmentation through random sampling and local feature aggregation. PointNeXt (Qian et al., 2022) introduced a scalable architecture suitable for various point cloud tasks. Despite these advancements, these methods typically demand substantial annotated data for training, limiting their practical applications. Moreover, they primarily focus on fully supervised scenarios, lacking adaptability to novel categories or data-scarce situations.

**Few-shot Point Cloud Semantic Segmentation**. To address the data scarcity challenge in point cloud semantic segmentation, few-shot learning approaches (Snell et al., 2017) have gained significant attention. Zhao et al. (Zhao et al., 2021b) pioneered the application of prototype networks to this domain, proposing the AttMPTI method. Subsequent research primarily focused on feature enhancement, prototype optimization, and domain adaptation. Notable contributions include the Bidirectional Feature Globalization (BFG) method by Mao et al. (Mao et al., 2022), which improved performance through feature interaction between support and query sets, and the Transformer-based SCAT method by Zhang et al. (Zhang et al., 2023a), which utilized hierarchical attention mechanisms to capture long-range dependencies. He et al. (He et al., 2023) introduced prototype adaptation and projection techniques to optimize prototype representations, while Xu et al. (Xu et al., 2023) proposed a robust few-shot segmentation framework to enhance domain adaptation capabilities. Although these methods have shown improvements, they still face challenges in effectively capturing complex local geometric structures and addressing domain differences, necessitating more effective solutions.

**Dynamic Convolution**. Dynamic convolution enhances a model's adaptability and expressive power by generating convolution kernels dynamically based on input data. In 2D image processing, CondConv (Yang et al., 2019) implemented dynamic convolution by combining multiple expert filters, significantly improving model performance without substantially increasing parameter count. ODConv (Li et al., 2022) further refined this approach, enabling dynamic adjustments across spatial, channel, and filter dimensions. Inspired by these successes, researchers have extended dynamic convolution to 3D point cloud processing. DyCo3D (He et al., 2021) introduced dynamic context learning to better capture local point cloud features. KPConv (Thomas et al., 2019) generated dynamic convolution kernels by learning local geometric structures, while PAConv (Xu et al., 2021) introduced a weight bank and ScoreNet to dynamically assemble convolution kernels, adapting to the irregular structure of point clouds. However, existing 3D dynamic convolution methods primarily rely on combining multiple convolution kernels through attention coefficients, leaving room for improvement in fine-grained semantic understanding of local geometric structures.

## 3 Method

In this section, we first introduce the definition of few-shot 3D point cloud semantic segmentation. Then, we present the definitions of Taylor series and dynamic convolution. Next, we introduce our proposed dynamic Taylor convolution. Following that, we describe our interactive prototype refinement module. Finally, we present the dynamic Taylor convolutional neural network (see Fig. 2) built upon the dynamic Taylor convolution and interactive prototype refinement module.

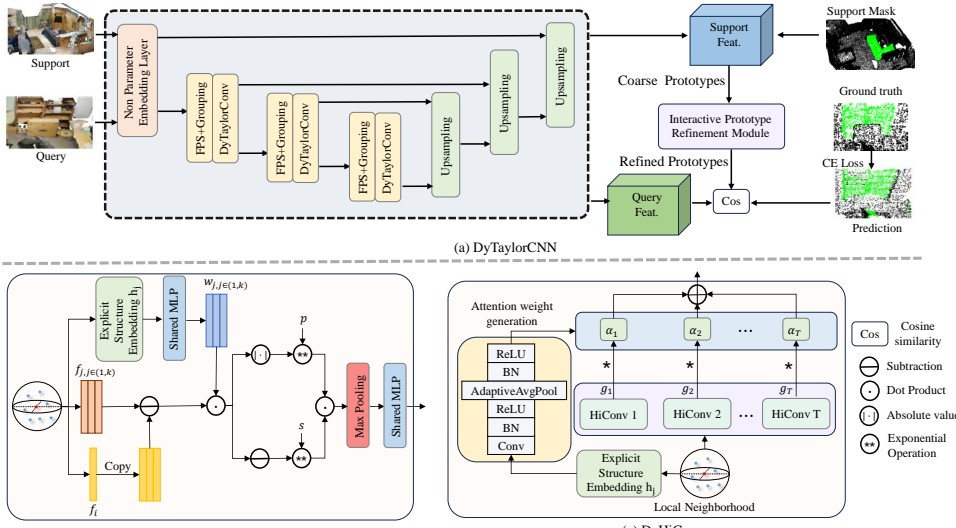

Figure 2: The architecture of DyTaylorCNN for few-shot point cloud semantic segmentation. (a) The backbone network centers around DyTaylorConv, a novel local feature extraction module inspired by the Taylor series. DyTaylorConv combines with FPS and Grouping to form the DyTaylor Block, which is stacked with upsampling operations to construct the encoder-decoder structure. (b) The Interactive Prototype Refinement (IPR) module, designed to reduce the feature distribution discrepancy between query and support sets. It consists of two key components: the Prototype Enhancement Module (PEM) and the Prototype Refinement Module (PRM). The IPR module is parameter-efficient and can be easily integrated into other few-shot learning frameworks as a plug-and-play component.

## 3.1 Taylor Series and Dynamic Convolution

**Taylor Series** (Rudin et al., 1964) is a local polynomial approximation of a function, allowing precise representation of a function near a given point with a finite number of polynomial terms. For a smooth function $f(x)$ at position $x_0$, the Taylor series expansion at point $x_0$ can be expressed as:

$$f(x) \approx f(x_0) + \sum_{n=1}^{\infty} \frac{f^{(n)}(x_0)}{n!}(x - x_0)^n, \tag{1}$$

where $f^{(n)}(x_0)$ represents the $n$-th order derivative of $f(x)$ at $x_0$, and $n$ is the order of expansion.

**Dynamic Convolution** (Yang et al., 2019) aims to enhance the modeling capability of networks by dynamically generating convolution kernels based on input data. Taking the dynamic convolution used in PAConv (Xu et al., 2021) as an example, for a point cloud with coordinates $\mathcal{P} = \{p_i | i = 1, \ldots, N\} \in \mathbb{R}^{N \times 3}$ and corresponding input features $\mathcal{F} = \{f_i | i = 1, \ldots, N\} \in \mathbb{R}^{N \times C_{in}}$, the output features after dynamic convolution are $\mathcal{G} = \{g_i | i = 1, \ldots, N\} \in \mathbb{R}^{N \times C_{out}}$, where $C_{in}$ and $C_{out}$ are the input and output feature channel numbers. Thus, convolution in the point cloud domain can be expressed as:

$$g_i = \mathcal{A}(\{w(p_j)f_j | p_j \in \mathcal{N}(p_i)\}), \tag{2}$$

where $\mathcal{A}$ represents the aggregation function, typically max pooling, average pooling, or summation. $\mathcal{N}(p_i)$ and $p_j$ denote the local neighborhood of the center point $p_i$ and neighboring points, respectively. $w(p_j)$ represents the weight corresponding to the feature $f_j$ of $p_j$. In dynamic convolution, $w(p_j)$ is usually composed of multiple weights $w_t(p_i)$, with attention coefficients $\alpha_t(p_i)$ generated for each weight in a data-driven manner:

$$w(p_j) = \sum_{t=1}^{T} \alpha_t(p_j) \odot w_t(p_j), \tag{3}$$

where $\odot$ denotes element-wise multiplication. Dynamic convolution in the point cloud domain can be expressed as:

$$g_i = \mathcal{A}(\{\sum_{t=1}^{T}(\alpha_t(p_j) \odot w_t(p_j))f_j | p_j \in \mathcal{N}(p_i)\}). \tag{4}$$

By comparing Eq. 4 and Eq. 1, we can observe that dynamic convolution can be viewed as a simplified version of Taylor series in representing local point cloud structures. However, it overlooks the description of high-order term features for local point cloud details and the importance of relative features in local structures.

### 3.2 Dynamic Taylor Convolution

Inspired by Taylor series (Rudin et al., 1964) and dynamic convolution (Yang et al., 2019), we designed a novel Dynamic Taylor Convolution (DyTaylorConv) by combining their strengths to capture local geometric structures more precisely. The dynamic Taylor convolution consists of two parts: Low-order Convolution (LoConv) and Dynamic High-order Convolution (DyHiConv). DyTaylorConv can be expressed as:

$$g_i = g_i^L + g_i^{DH}, \tag{5}$$

where $g_i^L$ and $g_i^{DH}$ represent the outputs of LoConv and DyHiConv, respectively.

**Low-order Convolution**. We adopt the design of PointNN (Zhang et al., 2023b), utilizing Nonparameterless Trigonometric Functions (NTF) to encode basic local structural information. First, we map the point cloud coordinates $p_i$ and color information $c_i \in \mathbb{R}^3$ to the same dimension as their features, then add their information and apply a non-linear transformation to obtain a high-dimensional representation of basic structural information. The LoConv can be formulated as:

$$g_i^L = \mathcal{A}(\{\mathbf{W}_l f_j' | p_j \in \mathcal{N}(p_i)\}), \tag{6}$$

$$f_j' = (f_j^p + f_j^c + f_j)/3, \tag{7}$$

$$f_j^p = [sin(\alpha p_j/\beta^{\frac{6i}{d}}), cos(\alpha p_j/\beta^{\frac{6i}{d}})] \in \mathbb{R}^d, \quad i = 1, \cdots, d, \tag{8}$$

where $f_j^c$ is obtained similarly to $f_j^p$. $\alpha$ and $\beta$ represent the wavelength and amplitude hyperparameters of the trigonometric functions, respectively. $\mathbf{W}_l \in \mathbb{R}^{C_{in} \times C_{out}}$ denotes the non-linear transformation matrix.

**Dynamic High-order Convolution**. To capture the details of complex local geometric structures, we draw inspiration from dynamic convolution (Yang et al., 2019) to generate multiple convolution weights using input information. Borrowing from the Taylor series concept, we use different orders of neighboring points $p_j$ and center point $p_i$ to capture different levels of information in local structures. Thus, DyHiConv (see Fig. 3(c)) can be expressed as:

$$g_i^{DH} = \phi_1 g_i^1 + \phi_2 g_i^2 + \cdots + \phi_N g_i^V, \tag{9}$$

where $g_i^v = \mathcal{A}(\mathcal{T}(f_i, f_j) | p_j \in \mathcal{N}(p_i))$ represents High-order Convolution(HiConv), $V$ is the number of HiConv, and $\mathcal{T}(f_i, f_j) = (\frac{w_j \odot (f_j - f_i)}{|w_j \odot (f_j - f_i)|})^s \odot |w_j \odot (f_j - f_i)|^p$ is a novel affine basis

function we designed, which can simulate the high-order terms of Taylor series and is called a high-order neuron. Here, $|\odot|$ represents element-wise absolute value, $s \in \{0, 1\}$, and $p$ is a learnable parameter.

$\phi_n$ represents the attention assembly coefficient of HiConv, which is also constructed from explicit geometric information $h_j$, specifically:

$$\phi_v = \frac{exp(\mathbf{W}_v h_j)}{\sum_{t=1}^{V} exp(\mathbf{W}_t h_j)}, \tag{10}$$

where $\mathbf{W}_v \in \mathbb{R}^{10 \times 1}$ denotes the non-linear transformation matrix.

**Explicit Structure Introduction**. We use the coordinates of neighboring points $p_j$ and center point $p_i$ as basic geometric elements to construct the weight $w_j$ for HiConv, which can be expressed as $w_j = \mathbf{W}_h h_j$. Here, $h_j = [p_i, p_j, p_j - p_i, \|p_i, p_j\|] \in \mathbb{R}^{10}$, and $\mathbf{W}_h \in \mathbb{R}^{10 \times C_{out}}$ denotes the non-linear transformation matrix. The introduction of explicit geometric information facilitates the learning of relative spatial layout relationships between points and the capture of local geometric features and details.

### 3.3 Interactive Prototype Refinement Module

Due to the significant domain difference in feature distribution between support and query sets, directly using prototypes generated from the support set for segmentation may lead to performance degradation. To address this issue, we propose the Interactive Prototype Refinement (IPR) Module (see Fig. 3), which consists of two key components: Prototype Enhancement Module (PEM) and Prototype Refinement Module (PRM), which transforms coarse prototypes into fine-graine d prototypes through two sub-modules: prototype enhancement and prototype refinement.

First, we perform local max pooling and mapping along the point dimension of support features $F_s \in \mathbb{R}^{M \times C}$ and query features $F_q \in \mathbb{R}^{M \times C}$ separately to learn the statistical characteristics of each channel. We also further map the prototype features $F_p \in \mathbb{R}^{(K+1) \times C}$ to increase their flexibility. The specific formulas are as follows:

$$F_s = MaxPooling(F_s) \times W_1 \in \mathbb{R}^{M' \times C}, \tag{11}$$

$$F_q = MaxPooling(F_q) \times W_1 \in \mathbb{R}^{M' \times C}, \tag{12}$$

$$F_p = F_p \times W_2 \in \mathbb{R}^{K \times C}, \tag{13}$$

where $W_1 \in \mathbb{R}^{C \times C}$ and $W_2 \in \mathbb{R}^{C \times C}$ represent learnable non-linear transformation matrices, and $MaxPooling$ denotes local max pooling.

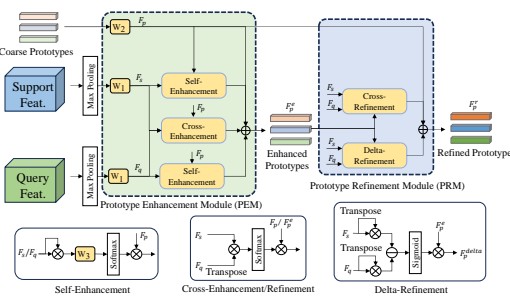

Figure 3: IPR module. It consists of two key components: Prototype Enhancement Module (PEM) and Prototype Refinement Module (PRM). IPR effectively reduces the feature distribution discrepancy between query and support sets, enhancing the model's few-shot learning capability. This module is parameter-efficient and can be easily integrated into various few-shot learning frameworks.

**Prototype Enhancement Module**. First, we learn self-enhancement attention coefficients from the features of query set and support set separately, i.e., $A_s = W_3(F_s^T F_s) \in \mathbb{R}^{C \times C}$ and $A_q = W_3(F_q^T F_q) \in \mathbb{R}^{C \times C}$, then obtain the updated prototype features $F_p^{self} = Softmax(A_s)F_p + Softmax(A_q)F_p$, where $W_3 \in \mathbb{R}^{C \times C}$ represents a learnable non-linear transformation matrix. Then we obtain the enhanced prototype features $F_p^{cross} = Softmax(A_{cross}) \odot F_p$ by learning the mutual information of support set and

query set, where $A_{cross} = F_q^T F_s \in \mathbb{R}^{C \times C}$. Therefore, the prototype features output by the PEM are as follows:

$$F_p^e = F_p^{self} + F_p^{cross} + F_p. \tag{14}$$

**Prototype Refinement Module**. To further utilize the difference between query set and support set features to refine the prototypes, we calculate the delta-refinement degree of the feature difference between query set and support set to alleviate domain bias, i.e., $\triangle_G = F_q^T F_q - F_s^T F_s$, then obtain $F_p^{delta} = sigmoid(\triangle_G) \odot F_p^e$. Furthermore, we obtain the enhanced prototype features $F_p^{e\_cross} = Softmax(A_{cross}) \odot F_p^e$ by cross-refinement operation, where $A_{cross} = F_q^T F_s \in \mathbb{R}^{C \times C}$. Therefore, the PRM are as follows:

$$F_p^r = F_p^{delta} + F_p^{e\_cross} + F_p^e. \tag{15}$$

The output of the PRM serves as the fine-grained prototype of the IPR module for matching with query features.

## 4 Experiments

For information about the architecture and experimental details of DyTaylorCNN, please refer to the Appendix A.

### 4.1 Datasets and Evaluation Metrics

We evaluate DyTaylorCNN on two widely used 3D point cloud datasets: S3DIS (Armeni et al., 2016) and ScanNet (Dai et al., 2017).

To conduct few-shot learning experiments, we divide the categories of each dataset into two non-overlapping subsets, denoted as $S_0$ and $S_1$. When one subset is designated as the test set, the other subset serves as the training set.

**Evaluation Metric** : We adopt the mean Intersection over Union (mIoU), a widely used metric for point cloud segmentation tasks, as our performance evaluation metric.

### 4.2 Comparison with existing methods

To evaluate our method, we compared it with DGCNN (Wang et al., 2019), ProtoNet (?), MPTI (Zhao et al., 2021b), AttMPTI (Zhao et al., 2021b), BFG (Mao et al., 2022), 2CBR (Zhu et al., 2023), PAP3D (He et al., 2023), Seg-PN (Zhu et al., 2024).

**Results analysis on the S3DIS dataset**. On the S3DIS dataset, DyTaylorCNN demonstrated exceptional performance. As shown in Table 1, in the 2-way 1-shot setting, DyTaylorCNN achieved an average mIoU of 71.95%, surpassing the previous best method Seg-PN (Zhu et al., 2024) by 5.54 percentage points. In the more challenging 3-way 1-shot setting, DyTaylorCNN reached an average mIoU of 66.26%, exceeding Seg-PN (Zhu et al., 2024) by 6.49 percentage points. These results highlight DyTaylorCNN's powerful feature extraction and generalization capabilities even with extremely limited labeled data. Notably, DyTaylorCNN maintained the best performance across all settings, including 5-shot scenarios, demonstrating the method's consistency and stability. These significant improvements indicate that DyTaylorCNN can more effectively capture local geometric features of point clouds and excel in few-shot learning tasks.

**Results analysis on the ScanNet dataset**. DyTaylorCNN also exhibited impressive performance on the ScanNet dataset. As illustrated in Table 2, in the 2-way 1-shot setting, DyTaylorCNN achieved an average mIoU of 71.96%, outperforming Seg-PN (Zhu et al., 2024) by 8.22 percentage points. In the 3-way 1-shot setting, DyTaylorCNN attained an average mIoU of 70.97%, surpassing Seg-PN (Zhu et al., 2024) by 7.40 percentage points. Particularly noteworthy is the 3-way 5-shot setting, where DyTaylorCNN achieved an average mIoU of 72.65%, exceeding Seg-PN (Zhu et al., 2024) by 7.05 percentage points, one of

Table 1: Few-shot Results (%) on S3DIS. $S_i$ denotes the split $i$ is used for testing. Avg is their average mIoU. The best results are shown in bold. The underline indicates the second best result.

| Method | Param. | 2-way | | | | | | 3-way | | | | | |
| | | 1-shot | | | 5-shot | | | 1-shot | | | 5-shot | | |
| | | $S_0$ | $S_1$ | Avg | $S_0$ | $S_1$ | Avg | $S_0$ | $S_1$ | Avg | $S_0$ | $S_1$ | Avg |
|---|---|---|---|---|---|---|---|---|---|---|---|---|---|
| DGCNN | 0.62 M | 36.34 | 38.79 | 37.57 | 56.49 | 56.99 | 56.74 | 30.05 | 32.19 | 31.12 | 46.88 | 47.57 | 47.23 |
| ProtoNet | 0.27 M | 48.39 | 49.98 | 49.19 | 57.34 | 63.22 | 60.28 | 40.81 | 45.07 | 42.94 | 49.05 | 53.42 | 51.24 |
| MPTI | 0.29 M | 52.27 | 51.48 | 51.88 | 58.93 | 60.56 | 59.75 | 44.27 | 46.92 | 45.60 | 51.74 | 48.57 | 50.16 |
| AttMPTI | 0.37 M | 53.77 | 55.94 | 54.86 | 61.67 | 67.02 | 64.35 | 45.18 | 49.27 | 47.23 | 54.92 | 56.79 | 55.86 |
| BFG | - | 55.60 | 55.98 | 55.79 | 63.71 | 66.62 | 65.17 | 46.18 | 48.36 | 47.27 | 55.05 | 57.80 | 56.43 |
| 2CBR | 0.35 M | 55.89 | 61.99 | 58.94 | 63.55 | 67.51 | 65.53 | 46.51 | 53.91 | 50.21 | 55.51 | 58.07 | 56.79 |
| PAP3D | 2.45 M | 59.45 | 66.08 | 62.76 | 65.40 | 70.30 | 67.85 | 48.99 | 56.57 | 52.78 | 61.27 | 60.81 | 61.04 |
| Seg-PN | 0.24 M | 64.84 | 67.98 | 66.41 | 67.63 | 71.48 | 69.56 | 59.11 | 60.42 | 59.77 | 59.48 | 64.72 | 62.10 |
| **DyTaylorCNN** | 0.68 M | **71.17** | **72.70** | **71.95** | **71.57** | **74.10** | **72.84** | **63.52** | **68.99** | **66.26** | **66.72** | **69.64** | **68.18** |
| Improvement | - | +6.33 | +4.72 | +5.54 | +3.94 | +2.62 | +3.28 | +4.41 | +8.57 | +7.24 | +7.24 | +4.92 | +6.08 |

Table 2: Few-shot Results (%) on ScanNet. $S^i$ denotes the split $i$ is used for testing. Avg is their average mIoU. The best results are shown in **bold**. The underline indicates the second best result.

| Method | Param. | 2-way | | | | | | 3-way | | | | | |
| | | 1-shot | | | 5-shot | | | 1-shot | | | 5-shot | | |
| | | $S_0$ | $S_1$ | Avg | $S_0$ | $S_1$ | Avg | $S_0$ | $S_1$ | Avg | $S_0$ | $S_1$ | Avg |
|---|---|---|---|---|---|---|---|---|---|---|---|---|---|
| DGCNN | 1.43 M | 31.55 | 28.94 | 30.25 | 42.71 | 37.24 | 39.98 | 23.99 | 19.10 | 21.55 | 34.93 | 28.10 | 31.52 |
| ProtoNet | 0.27 M | 33.92 | 30.95 | 32.44 | 45.34 | 42.01 | 43.68 | 28.47 | 26.13 | 27.30 | 37.36 | 34.98 | 36.17 |
| MPTI | 0.29 M | 39.27 | 36.14 | 37.71 | 46.90 | 43.59 | 45.25 | 29.96 | 27.26 | 28.61 | 38.14 | 34.36 | 36.25 |
| AttMPTI | 0.37 M | 42.55 | 40.83 | 41.69 | 54.00 | 50.32 | 52.16 | 35.23 | 30.72 | 32.98 | 46.74 | 40.80 | 43.77 |
| BFG | - | 42.15 | 40.52 | 41.34 | 51.23 | 49.39 | 50.31 | 34.12 | 31.98 | 33.05 | 46.25 | 41.38 | 43.82 |
| 2CBR | 0.35 M | 50.73 | 47.66 | 49.20 | 52.35 | 47.14 | 49.75 | 47.00 | 46.36 | 46.68 | 45.06 | 39.47 | 42.27 |
| PAP3D | 2.45 M | 57.08 | 55.94 | 56.51 | 64.55 | 59.64 | 62.10 | 55.27 | 55.60 | 55.44 | 59.02 | 53.16 | 56.09 |
| Seg-PN | 0.24 M | 63.15 | 64.32 | 63.74 | 67.08 | 69.05 | 68.07 | 61.80 | 65.34 | 63.57 | 62.94 | 68.26 | 65.60 |
| **DyTaylorCNN** | 0.68 M | **71.07** | **72.84** | **71.96** | **72.63** | **74.48** | **73.56** | **69.76** | **72.17** | **70.97** | **72.97** | **72.33** | **72.65** |
| Improvement | - | +7.92 | +8.52 | +8.22 | +5.55 | +5.43 | +5.49 | +7.96 | +6.83 | +7.40 | +10.03 | +4.07 | +7.05 |

the largest improvements across all settings. These results not only demonstrate DyTaylor-CNN's excellent performance across different datasets but also showcase its superior ability in handling more complex scenarios and effectively utilizing additional samples. DyTaylor-CNN's outstanding performance on the ScanNet dataset further validates its effectiveness and generalization capability in point cloud few-shot semantic segmentation tasks.

### 4.3 Ablation experiments

#### 4.3.1 Ablation experiments of DyHiConv

Table 3a demonstrates a clear improvement trend as we increase HiConv from 1 to 8 in the 2-way-1-shot setting on the S3DIS dataset. Starting with a single HiConv, the model achieves an average mIoU of 70.10%, consistently improving to a peak of 71.95% with 8 HiConv. The most significant improvement occurs between 1 and 2 convolutions, with a 0.73% increase in average mIoU, suggesting that even one additional HiConv significantly enhances the model's ability to capture complex local geometric features. However, the improvement rate gradually decreases beyond 4 convolutions, indicating a potential saturation effect. The marginal gain from 6 to 8 convolutions is only 0.42%, implying diminishing returns. This trend suggests that DyTaylorCNN effectively leverages the increased representational power of multiple HiConv to better model intricate geometric relationships in point cloud.

#### 4.3.2 Ablation experiments of explicit structure $h_j$

Table 3b shows a consistent improvement as more geometric information is incorporated into $h_j$. With only neighboring point coordinates $[p_j]$, the model achieves an average mIoU of 70.70%. Adding center point coordinates $[p_i, p_j]$ increases performance to 71.32%, suggesting the importance of relative positioning. Including relative displacement $[p_i, p_j, p_j - p_i]$ further improves mIoU to 71.77%, indicating that explicit spatial relationships benefit local

|  | (a) | | | | | (b) | | | |
|---|---|---|---|---|---|---|---|---|---|

| Number | 2-way-1-shot | | | | Number | 2-way-1-shot | | |
|---|---|---|---|---|---|---|---|---|
|  | $S_0$ | $S_1$ | Avg | |  | $S_0$ | $S_1$ | Avg |
| 1 | 68.03 | 72.16 | 70.10 | | $[p_j]$ | 69.55 | 71.84 | 70.70 |
| 2 | 69.07 | 72.58 | 70.83 | | $[p_i, p_j]$ | 70.42 | 72.21 | 71.32 |
| 4 | 70.09 | 72.42 | 71.26 | | $[p_i, p_j, p_j - p_i]$ | 70.98 | 72.56 | 71.77 |
| 6 | 70.41 | 72.64 | 71.53 | | $[p_i, p_j, p_j - p_i, \|p_i, p_j\|]$ | 71.17 | 72.70 | 71.95 |
| 8 | 71.17 | 72.70 | 71.95 | | | | | |

Table 3: (a) Effect of the number of HiConv on DyTaylorCNN. We report the results (%) under 2-way-1-shot settings on S3DIS datasets. (b) Effect of explicit structure $h_j$ on DyTaylorCNN. We report the results (%) under 2-way-1-shot settings on S3DIS datasets.

|  | (a) | | | | | | (b) | | |
|---|---|---|---|---|---|---|---|---|---|

| Number | 2-way-1-shot | | | | PEM | PRM | 2-way-1-shot | | |
|---|---|---|---|---|---|---|---|---|---|
|  | $S_0$ | $S_1$ | Avg | |  |  | $S_0$ | $S_1$ | Avg |
| ABF | 70.12 | 71.60 | 70.86 | | ✗ | ✗ | 48.78 | 51.81 | 50.30 |
| RBF | 62.64 | 63.11 | 62.88 | | ✓ | ✗ | 69.32 | 71.82 | 70.57 |
| s=0 | 69.56 | 70.96 | 70.26 | | ✗ | ✓ | 68.44 | 71.66 | 70.05 |
| s=1 | 71.17 | 72.70 | 71.95 | | ✓ | ✓ | 71.17 | 72.70 | 71.95 |

Table 4: (a) Effect of HiConv's Parameters on DyTaylorCNN. We report the results (%) under 2-way-1-shot settings on S3DIS datasets. (b) Effect of different composition of IPR on S3DIS under 2-way-1-shot settings on the $S_0$ and $S_1$ split.

structure understanding. The best performance (71.95% mIoU) is achieved with the most comprehensive representation $[p_i, p_j, p_j - p_i, |p_i, p_j|]$, which includes Euclidean distance between points. This configuration shows a 1.25% improvement over the baseline. These results underscore the significance of rich geometric feature representation.

### 4.3.3 Ablation experiments of HiConv

Table 4a demonstrate the significance of the parameter $s$ in shaping the model's effectiveness. The Affine Basis Function (ABF) configuration achieves a respectable average mIoU of 70.86%, indicating its capability in capturing local geometric features. However, the Radial Basis Function (RBF) setup performs notably worse, with an average mIoU of 62.88%, suggesting its limited ability to model complex point cloud structures in this context. Setting $s = 0$ yields an average mIoU of 70.26%, which is competitive but not optimal. The best performance is achieved when $s = 1$, resulting in an average mIoU of 71.95%. This configuration outperforms all others, demonstrating a 1.09% improvement over ABF and a substantial 9.07% gain over RBF. These results highlight the importance of the directional component in HiConv when $s = 1$. This setting allows the model to capture both magnitude and direction information in local point neighborhoods.

### 4.3.4 Ablation experiments of IPR

Table 4b presents the ablation study of the Interactive Prototype Refinement (IPR) module, comprising the Prototype Enhancement Module (PEM) and Prototype Refinement Module (PRM). Without PEM or PRM, the model achieves an average mIoU of 50.30%. Introducing PEM alone significantly boosts performance to 70.57%, a 20.27% improvement, underscoring PEM's crucial role in enhancing prototype features. PRM alone yields a slightly lower but still significant improvement, reaching 70.05% mIoU. This suggests PRM effectively refines prototypes, albeit less effectively than PEM in isolation. The full IPR module, combining both PEM and PRM, achieves the best performance with 71.95% mIoU, surpassing individ-

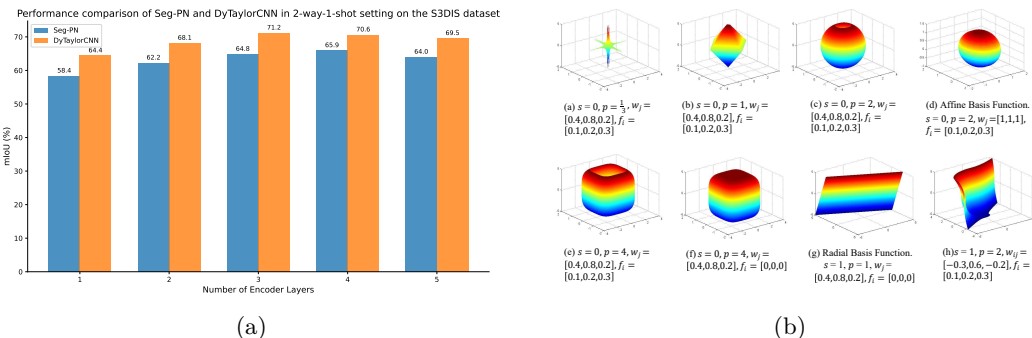

Figure 4: (a) Ablation for Number of Encoder Layers in 2-way-1-shot setting on the S3DIS dataset. (b) The visualization of different shapes of HiConv.

ual submodule performances by 1.38% and 1.90% respectively. This synergy indicates that PEM and PRM complement each other.

### 4.3.5 Ablation experiments of different numbers of Encoder Layers

Figure 4a illustrates the performance of Seg-PN and DyTaylorCNN across different numbers of encoder layers on the S3DIS dataset. DyTaylorCNN achieves its peak performance with three encoder layers, reaching an mIoU of 71.17%. This suggests that three layers provide an optimal balance between feature learning capacity and model complexity for DyTaylorCNN. The performance slightly decreases with four layers (70.61% mIoU) and further declines with five layers (69.54% mIoU), possibly due to overfitting or the vanishing gradient problem in deeper networks.Seg-PN shows a similar trend, but it also fails to achieve the high performance levels of DyTaylorCNN. This consistent superiority highlights DyTaylorCNN's more effective architecture for capturing local point cloud structures.

### 4.4 Visualization of Geometric Structure of HiConv

Fig. 4b illustrates the geometric versatility of HiConv under various parameter settings. When $s = 0$, $p = 1$ and $f_i = 0$, HiConv functions as an Affine Basis Function (ABF), representing a hyperplane (Fig. 4b(g)). With $s = 0$ and $p = 2$, it becomes a Radial Basis Function (RBF), forming an isotropic closed hypersphere (Fig. 4b(d)). The parameter $s$ influences output feature directionality and Signed Cosine Power (HiConv) closure, while $p$ determines neuron morphology. As $p$ increases from $1/3$ to $4$ (Figs. 4b(a-c, e)), the shape evolves from concave to convex. Fig. 4b(f) shows the neuron's response when $f_i = 0$, and Fig. 4b(h) demonstrates HiConv's ability to model complex, asymmetric relationships. This adaptability allows HiConv to flexibly fit various geometric structures in point clouds, enabling more nuanced feature extraction for tasks like semantic segmentation by fine-tuning $s$ and $p$ and learning appropriate $w_j$ from local geometric priors.

## 5 Conclusion

This paper introduces the DyTaylorCNN for few-shot point cloud semantic segmentation, addressing domain gap and insufficient local geometric structure representation challenges. DyTaylorCNN's core innovations lie in the DyTaylorConv and the IPR module. DyTaylorConv captures geometric features through LoConv and DyHiConv, while the IPR Module reduces domain distribution gaps between support and query sets. Extensive experiments demonstrate our method's superior performance across various few-shot settings. Despite significant progress, limitations remain, such as the computational cost of power exponent operations in HiConv. Future work will focus on exploring more efficient model architectures, extending DyTaylorCNN to larger-scale point cloud, integrating with other modalities, and investigating potential applications in other point cloud understanding tasks.

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

## A    Appendix

### A.1    Problem Definition

In this study, we focus on the few-shot 3D point cloud semantic segmentation task, adopting the episodic learning paradigm to divide the dataset into seen classes $C_{seen}$ and unseen classes $C_{unseen}$. Each few-shot task is constructed as an N-way K-shot problem, comprising a support set $S$ and a query set $Q$. The support set $S$ consists of K labeled point cloud samples for each of the N categories, where each sample $P_s^{n,k} \in \mathbb{R}^{L \times (3+f_0)}$ is accompanied by its binary mask $M_s^{n,k} \in \mathbb{R}^{L \times 1}$. The query set $Q$ contains $H$ point cloud samples $P_q^i \in \mathbb{R}^{L \times (3+f_0)}$ to be segmented. $L$ represents the number of point clouds. $f_0$ represents the dimension of the point feature. Our objective is to leverage the limited annotations in the support set to accurately segment the query set point clouds, classifying them into N target classes and 1 background class. To achieve this, we transform the task into a point-level similarity matching problem, employing feature encoding, prototype generation, and similarity calculation to perform segmentation.

### A.2    The details of Dynamic Taylor Convolutional Neural Network

First, we adopt the SegNN (Zhu et al., 2024) approach to perform a parameterless high-dimensional mapping of the initial input features through trigonometric function encoding, mapping them to a high-dimensional space of dimension 60. Then we construct the dynamic Taylor Block with farthest point sampling-grouping-dynamic Taylor convolution as the basic component. We then stack three dynamic Taylor Blocks as the encoder of the dynamic Taylor convolutional neural network, with dimensions of 120, 240, 480 for each encoder. The decoder uses reverse interpolation algorithm to restore the resolution of the point cloud. Between the encoder and decoder, we adopt a skip connection structure similar to Unet, fully utilizing contextual information. The support set and query set pass through the encoder and decoder separately to obtain the corresponding features $F_s$ and $F_q$. Then, we use masked average pooling on the support set features $F_s$ to generate coarse prototypes $F_p$ for K+1 classes. Next, we pass these prototypes through the interactive prototype refinement module to obtain fine-grained prototype features $F_p^r$. Finally, we perform similarity matching between the query set features $F_q$ and $F_p^r$ to accurately segment points that the model has not seen before.

### A.3    Datesets

**S3DIS dataset** (Armeni et al., 2016) comprises 3D RGB point clouds from 272 rooms across 6 indoor environments. Each point is annotated with one of 13 semantic labels (12 semantic categories plus clutter). Following the setup in (Zhao et al., 2021b), we divide each point cloud scene into 1m × 1m blocks and randomly sample 2048 points from each block. The final S3DIS dataset is partitioned into 7547 blocks.

**ScanNet dataset** (Dai et al., 2017) contains a total of 1513 scanned scenes. All points, except for unannotated spaces, are labeled with one of 20 semantic categories. Following the processing method in (Zhao et al., 2021b), the ScanNet dataset is divided into 36350 blocks, each containing 2048 points.

Table 5 provides a list of class names for the $S_0$ and $S_1$ splits of both the S3DIS and ScanNet datasets.

Table 5: Seen and Unseen Classes Split for S3DIS and ScanNet. We follow Zhao et al. (2021b) to evenly assign categories to $S_0$ and $S_1$ splits.

|  | $S_0$ | $S_1$ |
|---|---|---|
| S3DIS | beam, board, bookcase, ceiling, chair, column | door, floor, sofa, table, wall, window |
| ScanNet | bathtub, bed, bookshelf, cabinet, chair, counter, curtain, desk, door, floor | other furniture, picture, refrigerator, show curtain, sink, sofa, table, toilet, wall, window |

A.4   Implementation Details

We implement our method using the PyTorch framework, with all experiments conducted on a single NVIDIA GeForce RTX 4090 GPU. The experiments are performed under the N-way K-shot setting, where N takes values from {2, 3} and K from {1, 5}. For each setting, we randomly sample 100 test episodes and report the average mIoU results. Training is conducted on the seen category set $C_{seen}$, while testing is performed on the unseen category set $C_{unseen}$ to evaluate the model's generalization capability.

For N-way K-shot tasks, we generate K prototypes for each category and use their average as the final prototype for that category. In the dynamic Taylor convolution, we set $\alpha = 2\pi$ and $\beta = 30$ for the LoConv, and initialize the learnable power exponent $p$ to 1 for the HiConv. The local neighborhood is constructed using the k-NN algorithm, selecting 16 nearest neighbors. In the IPR module, the stride for local max pooling is set to 32.

We adopt the episodic learning paradigm for training. In each training batch, we construct an episode containing a support set and a query set. The support set is randomly selected from N-way K-shot samples, while the query set is randomly drawn from N samples of unseen categories. To optimize model parameters, we employ the cross-entropy loss function to calculate the difference between query set predictions and ground truth labels. We use the AdamW optimizer ($\beta 1 = 0.9$, $\beta 2 = 0.999$) to update network parameters, with an initial learning rate of 0.001, which is halved every 7000 iterations.

A.5   The impact of the number of input point clouds

Table 6: Robustness of DyTaylorCNN with different number of point clouds.

| Number of points | 1024 | 2048 | 4096 | 6092 |
|---|---|---|---|---|
| DyTaylorCNN | 66.48 | 71.17 | 70.02 | 63.87 |

Table 6 demonstrates the performance variations of DyTaylorCNN under different point cloud densities. The model shows a notable increase in performance from 1024 to 2048 points, with mIoU improving from 66.48% to a peak of 71.17%. This significant boost suggests that DyTaylorCNN benefits from the increased geometric information provided by a moderate increase in point density. However, as the number of points further increases to 4096, there's a slight performance decline to 70.02%, indicating that the model maintains robust performance even with higher point densities. A more substantial drop is observed at 6092 points, where the mIoU decreases to 63.87%. This trend suggests that while Dy-TaylorCNN can effectively utilize additional point information up to a certain threshold, extremely high point densities may introduce challenges in feature extraction or increase

model complexity beyond optimal levels. The model's peak performance at 2048 points indicates an ideal balance between information richness and computational efficiency.

### A.6 Ablation Experiment on Hyperparameters of NTF

Table 7 presents the ablation study for the parameter $\beta$ in the Nonparameterless Trigonometric Functions (NTF) used in the LoConv of DyTaylorCNN. The results demonstrate the significant impact of $\beta$ on the model's performance. The model's performance exhibits a non-linear relationship with $\beta$. Starting from a lower value of 10, the mIoU increases as $\beta$ grows, reaching a peak of 71.17% at $\beta = 30$. This optimal value suggests that a moderate amplitude in the trigonometric functions provides the best encoding of local structural information.

Performance decreases for values both below and above the optimal $\beta = 30$. Lower values (10, 20) may result in insufficient feature discrimination, while higher values (40, 60) could lead to overfitting or loss of fine-grained details. Interestingly, very high $\beta$ values (80, 100) show a slight performance recovery, possibly due to the capture of larger-scale structures.

The 4.35% mIoU difference between the best ($\beta = 30$) and worst ($\beta = 10$) performances underscores the critical role of $\beta$ in NTF. This sensitivity highlights the importance of careful tuning for optimal point cloud feature encoding in DyTaylorCNN, balancing between local detail preservation and global structure capture.

Table 7: Ablation for Parameter $\beta$ in NTF.

| $\beta$ | 10 | 20 | 30 | 40 | 60 | 80 | 100 |
|---|---|---|---|---|---|---|---|
| DyTaylorCNN | 66.82 | 67.63 | 71.17 | 68.79 | 67.65 | 69.45 | 69.07 |

