# OpenReview forum: "Dynamic Taylor Convolutional Neural Network for Few-Shot Point Cloud Semantic Segmentation"
_ICLR.cc/2025/Conference — ICLR 2025 Conference Withdrawn Submission_

### Official Review · Reviewer_tKKT · 2024-10-19

**Soundness:** 3
**Presentation:** 3
**Contribution:** 2
**Rating:** 5
**Confidence:** 4

**Summary:**

The paper propose a  pre-training-free model for few-shot point cloud segmentation. The model leverage the Taylor series and dynamic convolution to model the low-order and high-order structure of the point cloud. A Interactive Prototype Refinement module is further proposed to mitigate the domain shift between the support and query sets.

**Strengths:**

The proposed method demonstrate state-of-the-art performances on S3DIS and ScanNet.

**Weaknesses:**

1. The motivation of this paper is similar to Seg-NN. In Seg-NN, they use Point-NN [R1] as encoder and also does not need pre-training stage, which make the contribution of this paper limited.
2. Using high-order Talyor series to model point cloud is also proposed in [R2], author should cite it and discuss the difference.
3. The design of IPR module is not well motivated. Since it is composed by several sub-modules, e.g. self-enhancement, cross-enhancement, and delta-refinment, however, the experiments does not prove the necessesties and demonstrate the soundness of these modules.
[R1] No Time to Train: Empowering Non-Parametric Networks for Few-shot 3D Scene Segmentation, CVPR2024
[R2] Surface Representation for Point Clouds, CVPR2022

**Questions:**

What is the pysical meaning of this affine base function ("T(fi,fj) in L269) used in HiConv? How to determine s?

---

### Official Review · Reviewer_8xLp · 2024-10-30

**Soundness:** 3
**Presentation:** 2
**Contribution:** 2
**Rating:** 5
**Confidence:** 4

**Summary:**

This paper aim to deal with few-shot point cloud segmentation. This paper propose a DyTaylorCNN for point cloud segmentation. They also introduce the Interactive Prototype Module to reduce domain gap. Experiments are conducted on S3DIS and ScanNet. Results indicate it shows better performance than competitors.

**Strengths:**

1. This paper shows better performance than the competitors.
2. Experiments on S3DIS and ScanNet indicates the robustness of the proposed method.

**Weaknesses:**

1. As the DyTaylerCNN is proposed to extract the basic geometric information of point clouds, it will be better to compare the feature extraction performance with PointTransformer v3 or Occupancy-like methods. As the compared baseline DGCNN is not the SOTA feature extraction method. Whether DyTalerCNN can be used in fully supervised segmentation.
2. The benefits of the Tayler series is not well illustrated
3. For the experiments, 2-way or 3-way settings are used. Whether such setting is enough to evaluate the performance. For the ScanNet, whether the original ScanNet is used or the ScanNet200. How the performance will be when the number of classes become larger.
4. As open-vocabulary point cloud segmentation method appears "Open-Vocabulary 3D Semantic Segmentation with Foundation Models" with competitive performance,  whether few-shot shows the advantage over open vocabulary segmentation.

**Questions:**

More reasonable experiment settings should be introduced to present the contribution and the advantage over open-vocabulary segmentation.

---

### Official Review · Reviewer_Fwf5 · 2024-11-01

**Soundness:** 3
**Presentation:** 3
**Contribution:** 3
**Rating:** 6
**Confidence:** 3

**Summary:**

The paper presents a new method based on dynamic taylor convolutional NN for few-shot point segmentation. To address the problem of few-shot point semantic segmentation, more specifically, the limitations of pretraining learning paradigm and insufficient local geometric structure representation, the paper presents DyTaylorCNN which includes Dynamic Taylor convolution and INteractive Prototype refinement module. Experimental results on the benchmark shows promising results of the proposed algorithm.

**Strengths:**

1. The problem of few-shot semantic segmentation is an challenging problem for the research community.
2. The proposed DyTaylorCNN has promising results on multiple benchmark datasets and sufficient ablation studies have been conducted to validate the effectiveness of the setting.

**Weaknesses:**

1. Rather than the comparison of parameters against the baselines, is it possible to provide the inference speed of the proposed algoithm compared with the existing approach in Table 1 and Table 2.

2. The baselines for comparisons in Table 1 and Table 2 may not be a fair comparison. For example, most of the publications are published two years ago with only one paper Seg-PN is published in 2024. By comparsing with Seg-PN, as the proposed paper has more than two times larger parameters, the comparison may not be fair. How about the performance of the proposed algorithm with similar parameters as Seg-PN?


3. Minor problems:
missign reference number for ProtoNet (?) in Page 7 Line 358

**Questions:**

Please address the problems raised in the weakness section.

---

### Official Review · Reviewer_rpML · 2024-11-08

**Soundness:** 2
**Presentation:** 2
**Contribution:** 2
**Rating:** 3
**Confidence:** 3

**Summary:**

This paper tackles the challenge of few-shot point cloud semantic segmentation by proposing DyTaylorCNN, a pre-training-free neural network that leverages the Taylor series and dynamic convolutions to enhance local structure representation. DyTaylorCNN introduces two modules: Dynamic Taylor Convolution (DyTaylorConv) and an Interactive Prototype Refinement (IPR) Module. DyTaylorConv combines Low-order Convolution (LoConv), which extracts basic geometric information, and Dynamic High-order Convolution (DyHiConv), which adapts to complex geometric structures using spatial priors. Although the proposed method with these modules shows impressive performance on few-shot point cloud segmentation benchmarks, there are a few concerns related to the novelty and comparison as described in the weakness section.

**Strengths:**

The model demonstrates impressive performance in few-shot point cloud segmentation, particularly on the ScanNet and S3DIS datasets. Additionally, visualizations aid in understanding the model’s mechanism and results.

**Weaknesses:**

- The performance gain of DyTaylorConv is not reported in the paper. Based on Tables 3-(a) and 4-(b), it seems that the performance gain mainly comes from Prototype Enhancement Module (PEM) and Prototype Refinement Module (PRM). Then, why do we need DyTaylorConv for few-shot 3D semantic segmentation? If DyTaylorConv is indeed effective, then I recommend the authors to evaluate its effectiveness on general 3D perception tasks (e.g., ScanNet semantic segmentation, ScanObjectNN shape classification) since it is not a specially designed for few-shot 3D semantic segmentation but a general operator. \
- Regarding to Low-order Convolution, if it is designed for capturing basic structural information, why don't you just use RepSurf [1]? I recommend the authors to compare Low-order Convolution with RepSurf.

[1] Ran et al., "Surface Representation for Point Clouds", CVPR, 2022.

**Questions:**

I have minor comments on paper writing. The paper contains a few typos, here are some of them:
- L243: Nonparameterless -> Nonparameter
- L263: Fig. 3(c) -> Fig. 2(c)
- L294: fine-graine d -> fine-grained

---

### Note · Authors · 2024-11-15

I have read and agree with the venue's withdrawal policy on behalf of myself and my co-authors.